# Cascading events during the 1650 tsunamigenic eruption of Kolumbo volcano

Jens Karstens [1]✉, Gareth J. Crutchley [1], Thor H. Hansteen [1], Jonas Preine [2], Steven Carey[3], Judith Elger [1], Michel Kühn [1], Paraskevi Nomikou[4], Florian Schmid[1,7], Giacomo Dalla Valle[5], Karim Kelfoun[6] & Christian Berndt [1]

Volcanic eruptions can trigger tsunamis, which may cause significant damage to coastal communities and infrastructure. Tsunami generation during volcanic eruptions is complex and often due to a combination of processes. The 1650 eruption of the Kolumbo submarine volcano triggered a tsunami causing major destruction on surrounding islands in the Aegean Sea. However, the source mechanisms behind the tsunami have been disputed due to difficulties in sampling and imaging submarine volcanoes. Here we show, based on three-dimensional seismic data, that ~1.2 km³ of Kolumbo's northwestern flank moved 500–1000 m downslope along a basal detachment surface. This movement is consistent with depressurization of the magma feeding system, causing a catastrophic explosion. Numerical tsunami simulations indicate that only the combination of flank movement followed by an explosive eruption can explain historical eyewitness accounts. This cascading sequence of natural hazards suggests that assessing submarine flank movements is critical for early warning of volcanogenic tsunamis.

Volcanogenic tsunamis have caused more than 55,000 fatalities since the late 18th century[1]. Tsunami generation is often due to a combination of processes, including submarine explosions, pyroclastic flows, caldera subsidence, and flank failures[2]. The 2022 Hunga Tonga-Hunga Ha'apai (HTHH) eruption in Tonga[3,4] and the 2018 Anak Krakatau sector collapse in Indonesia demonstrated the devastating tsunami hazard of volcanic eruptions in shallow marine environments and at the land-sea-boundary[5]. Both eruptions have become a focus of volcanological research in recent years. The 2018 tsunami of Anak Krakatau hit neighboring coasts without any warning and caused hundreds of casualties[5]. Satellite geodetic data from more than ten years before the event indicated deformation within the volcanic edifice[6]. However, it was only after the collapse that these observations were interpreted as revealing prolonged, deep-seated deformation in the buildup to the collapse[7]. Internal deformation of a volcanic flank preceding catastrophic failure has also been demonstrated for the 1888 Ritter Island sector collapse[8,9],

and episodic gravity-driven movement along deep detachment surfaces is known from several large volcanoes[10–12]. At Mt Etna, an average seaward motion of 3 to 5 mm per year is observed[11], while the southwestern flank of Kilauea shows both a transient cm-scale yearly seaward movement and m-scale slip events accompanied by major earthquakes[13].

Kolumbo is a fully submerged volcano 7 km northeast of Santorini and part of the Hellenic Volcanic Arc. It is by far the largest volcano of the Kolumbo Volcanic Chain, consisting of 24 volcanic cones formed ~350 kyrs ago[14,15] (Fig. 1). Kolumbo's edifice was created by at least five eruptive cycles, the earliest dating back more than 1 Myrs[15,16]. The most recent eruption, in 1650 CE, formed a cone consisting of up to ~260 m-thick stratified pumice deposits, which breached the sea surface before being destroyed by a violent explosive eruption that formed a 500-m-deep and 2500-m-wide crater[17,18]. The eruption deposits consist of highly vesicular rhyolitic pumice with cm-size mafic inclusions, likely reflecting an eruption triggered by mafic replenishment into a

[1]GEOMAR Helmholtz-Zentrum für Ozeanforschung Kiel, Kiel, Germany. [2]University of Hamburg, Institute of Geophysics, Hamburg, Germany. [3]University of Rhode Island, Kingston, USA. [4]National and Kapodistrian University of Athens, Athens, Greece. [5]Italian National Research Council, Institute of Marine Science ISMAR, Bologna, Italy. [6]Laboratoire Magmas et Volcans, Université Clermont Auvergne, OPGC, CNRS, IRD, F-63000 Clermont, Ferrand, France. [7]Present address: K.U.M Umwelt und Meerestechnik Kiel GmbH, Kiel, Germany. ✉e-mail: jkarstens@geomar.de

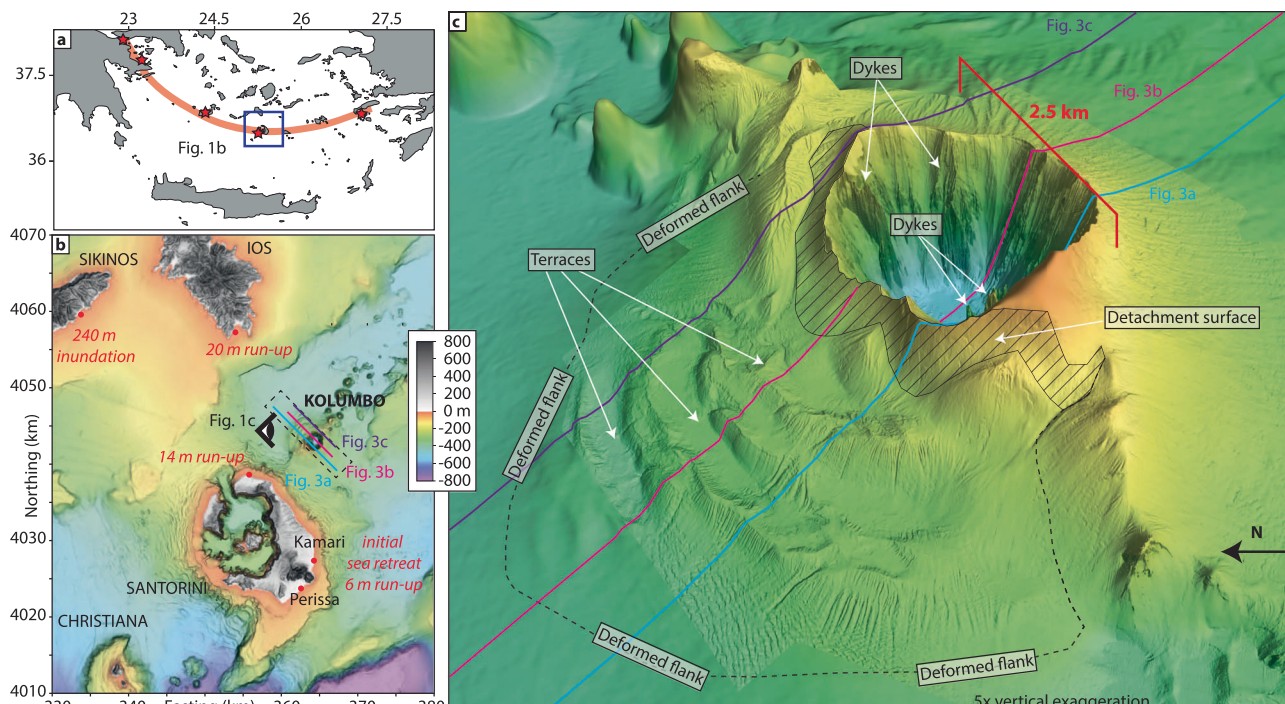

**Fig. 1 | Kolumbo volcano in the Aegean Sea. a** Map of the Hellenic Volcanic Arc (red) with active volcanoes (stars) and the location of this study area (blue box). **b** Topographic map of the Christiana-Santorini-Kolumbo volcanic zone and neighboring islands of Sikinos and Ios with historical tsunami eyewitness accounts. **c** Perspective view of high-resolution (2 m) bathymetry data[44] showing Kolumbo's deformed northwestern flank with terrace morphology, exposed part of the detachment surface, and the 1650 crater with morphological expression of dykes. Eye symbol in (**b**) shows the viewing direction. The cyan, magenta and purple lines indicate the position of the profiles depicted in Fig. 3.

cooler silicic reservoir[18,19]. The 1650 eruption triggered a tsunami causing major destruction on the neighboring islands of Santorini, Ios, and Sikinos. Historical eyewitness accounts reported maximum run-up heights of 20 m on the southern coast of Ios, 240 m inundation on Sikinos[20,21], and flooding of up to 2 km² of land on the eastern coast of Santorini which caused major destruction in the towns of Kamari and Perissa[17,20,21] (Fig. 1b). Flooding at Kamari and Perissa was preceded by sea retreat, indicating a negative-amplitude leading wave[20]. Onshore tsunami deposits have been identified up to 14 m above sea level on the north coast of Santorini, 6 m above sea level in Kamari, and up to 360 m inland in Perissa as well as on Ios and Sikinos[20,21].

The genesis of volcanogenic tsunamis is often complex and may involve underwater explosions, earthquakes, caldera subsidence, pyroclastic density currents, flank failures or a combination of these processes[2]. Even for prominent and comparably recent events, such as the 1883 eruption of Krakatau, the tsunami source mechanisms are still debated. Previous analyses of the 1650 tsunami tested the submarine emplacement of pyroclastic flows, caldera collapse and underwater explosions as potential singular source mechanisms[20].

However, the movement of Kolumbo's northwestern flank (Fig. 1c) has not been considered as a source mechanism in previous simulations. In this study, we use new high-resolution 3D reflection seismic data and numerical tsunami simulations to reconstruct the mechanisms that caused the tsunami waves recorded in 1650. We investigate Kolumbo's flank deformation and evaluate its tsunami-genic potential, as well as its role in triggering the violent explosions. Our analyses show that flank deformation initiated a cascade of events triggering the violent and tsunamigenic eruption of Kolumbo in 1650.

## Results

### Structural architecture of Kolumbo
High-resolution (2 m) bathymetry data reveal intricate morphological details of the cone formed by the 1650 eruption of Kolumbo (Fig. 1c).

The northwestern flank of the volcano is characterized by a distinctive terraced morphology, in stark contrast to the smooth flanks to the south and east (Fig. 1c). 3D seismic reflection data reveal the subsurface architecture of this structural asymmetry (Figs. 2 and 3). While the southeastern flank comprises undeformed, sub-parallel strata, the northwestern flank displays pronounced internal deformation (Figs. 2 and 3a). This deformed sedimentary unit lies directly above the ~1600 BCE Minoan eruption ignimbrites[16]. Aside from the 1650 eruption, there is no evidence of other recent explosive eruptions in sediment cores[22,23]. Therefore, the deformation affects solely volcanic sediments deposited by the 1650 eruption. Mapping of the top and base of the deformed sediments yields a volume of ~1.2 km³, assuming a seismic velocity of 1700 m/s (±50 m/s) based on refraction seismic data[15]. The bathymetric data show elongated ridges that strike slope parallel around the northwestern flank of the volcanic edifice (Fig. 1c). The analysis of the 3D seismic data reveals that they are the surface expression of anticlinal folds associated with downslope compression (Figs. 2 and 3). Their curved shapes in planform, their relative steepness, and interpreted internal thrusting indicate that the ridges are the result of compression and not contourite sediment depositional processes seen at some other submarine volcanoes[24]. Deformation at the toe of the flank is complex and segmented (manifesting itself in the terrace-like morphology), while a distinct detachment surface can be observed toward the top of the remnant cone (hatched region, Figs. 1c and 4a). The exposed detachment surface has a slope angle of ~19°, while the underlying strata dip at ~11°, which is significantly steeper than strata within the southwestern flank that dip at ~6° (Fig. 3a). The steeper stratigraphic dips beneath the northwestern flank are likely due to deposition onto a pre-existing remnant cone (Figs. 2 and 3a), which is not present beneath the southeastern flank. Additionally, there is a morphological step between the two slopes, as the base of the 1650 cone lies 50 - 100 m deeper on the northwestern side of the edifice (Fig. 3). The 1650 eruptive products were deposited on top of

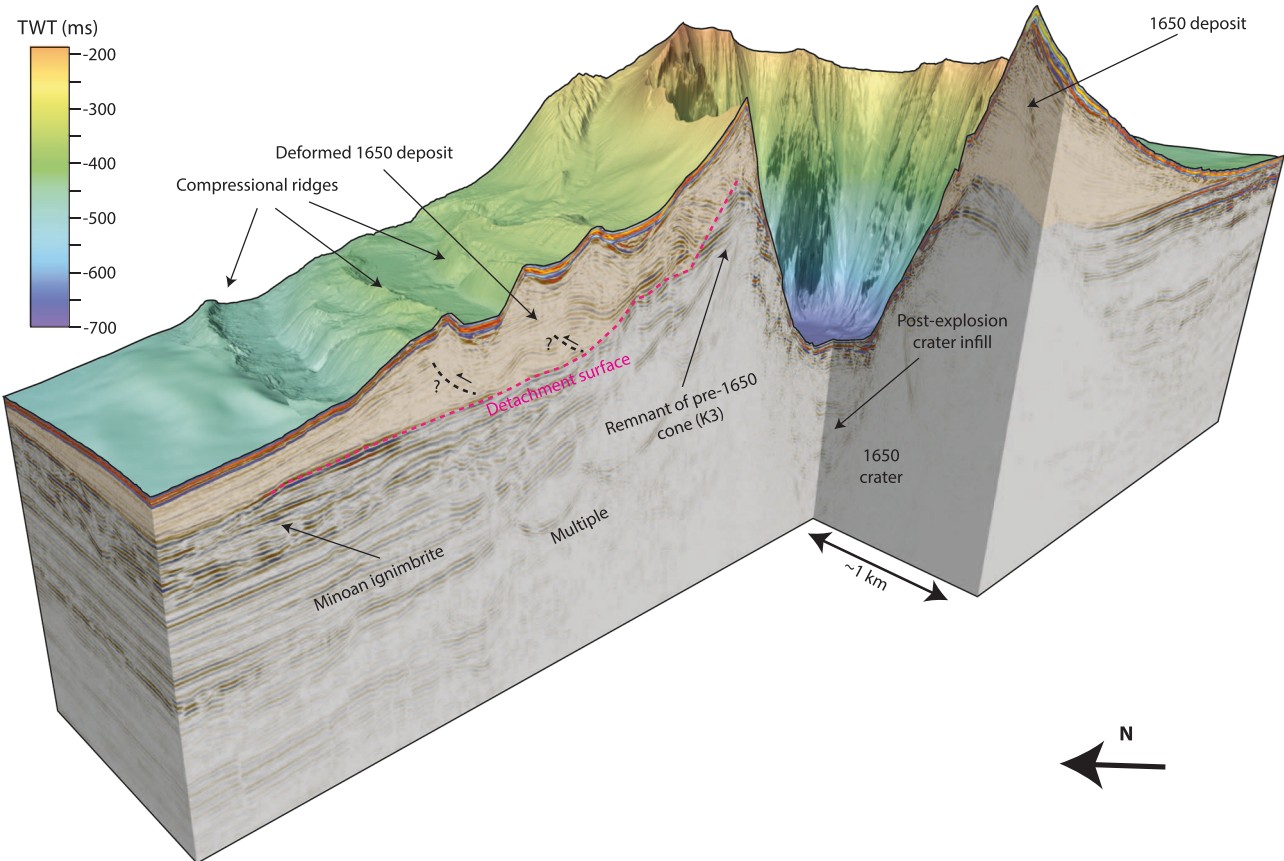

**Fig. 2 | 3D seismic dataset of the Kolumbo volcano.** Chair-cut view of the 3D seismic data around the volcanic edifice of Kolumbo, showing the deposits of the 1650 eruption (orange semi-transparent overlay) and the crater formed by an explosive eruption. The northwestern flank is characterized by deformation as the result of lateral displacement along a basal detachment surface (dashed pink line). Folded sediments above the detachment are consistent with compression. Dashed black lines are tentatively interpreted thrust faults associated with the seafloor ridges.

remnants of a cone from a previous eruption (K3 in ref. [15]), which introduce structural heterogeneities within the northwestern flank of Kolumbo, in contrast to the relatively flat southeastern flank (Fig. 3). This complex pre-eruption topography, with a steep northwestern flank, likely controlled the instability of this flank segment as slope angle is a dominant factor for slope stability[25]. Previous numerical simulations[20] suggested a violent explosion, pyroclastic flow emplacement, and caldera subsidence as potential tsunami source mechanisms during the 1650 eruption. However, the 3D seismic data show neither indications for sizable pyroclastic flow deposits (Figs. 2 and 3) nor for ring faults that would accommodate caldera collapse (Fig. 3b). Therefore, we deem pyroclastic flow emplacement and caldera collapse as unlikely tsunami source mechanisms. The existence of a 2500-m-wide and 500-m-deep crater, however, provides clear evidence for a highly explosive eruption phase. The seismic and bathymetric datasets suggest a link between this explosion and the flank deformation. At the top of the remnant cone, the detachment surface intersects prominent ridges that are the seafloor expression of feeder dykes (Fig. 4a, d). Their present exposure is probably due to their higher mechanical stability relative to the surrounding volcanoclastic deposits, which have been eroded since the 1650 eruption. While the explosion represents an obvious tsunami source mechanism, the previously unconsidered tsunamigenic potential of the flank deformation requires careful evaluation.

**Reassessment of the 1650 tsunami**

Historical eyewitness accounts of tsunamis can be used to test the feasibility of tsunami simulations. For Kolumbo, the observations of higher tsunami run-up at Ios than at Santorini and sea retreat preceding inundation in eastern Santorini[17,20,21] suggest a northwesterly directed wave propagation. Previous numerical simulations of the explosive eruption assumed a radial-symmetric initial waveform with a steep positive amplitude leading wave, followed by a trough that mimics the crater[20]. By assuming a peak wave amplitude of 250 m, these simulations could reproduce tsunami run-up heights at Santorini. However, the simulations could not reproduce the initial sea retreat at Kamari and Perissa, and they underestimated the 20 m run-up height at Ios by a factor of four. Therefore, it appears unlikely that an underwater explosion was the only source of the 1650 tsunami.

To evaluate the tsunami potential of flank deformation, we extrapolated the present-day southeastern flank slopes toward the sea surface to reconstruct a cone that just breaches the sea surface (Fig. 5) to be consistent with eyewitness accounts[17,18]. This pre-explosion topography (Fig. 5a) and the seismically-mapped detachment surface (Fig. 5b) were used as input surfaces for numerical tsunami simulations using VolcFlow[26]. Kolumbo's cone consists of poorly consolidated pumice with a median dry density of 0.725 g/cm³ and median vesicularity 67.5 (ref. [18]), which, depending on the inter-pore porosity, results in a bulk deposit density of 1250–1500 kg/m³. This density is significantly lower than values used in previous volcanic landslide tsunami simulations, e.g., 1900 kg/m³ at Anak Krakatau[5], 2000 kg/m³ at Ritter Island[27] and 2600 kg/m³ at Tenerife[28]. In VolcFlow simulations, the slide material density, in combination with its yield strength, controls the dynamics of flank deformation. We tested a large parameter space (Supplementary Information) with yield strengths between 2000 and 75,000 Pa and densities between 1250 and

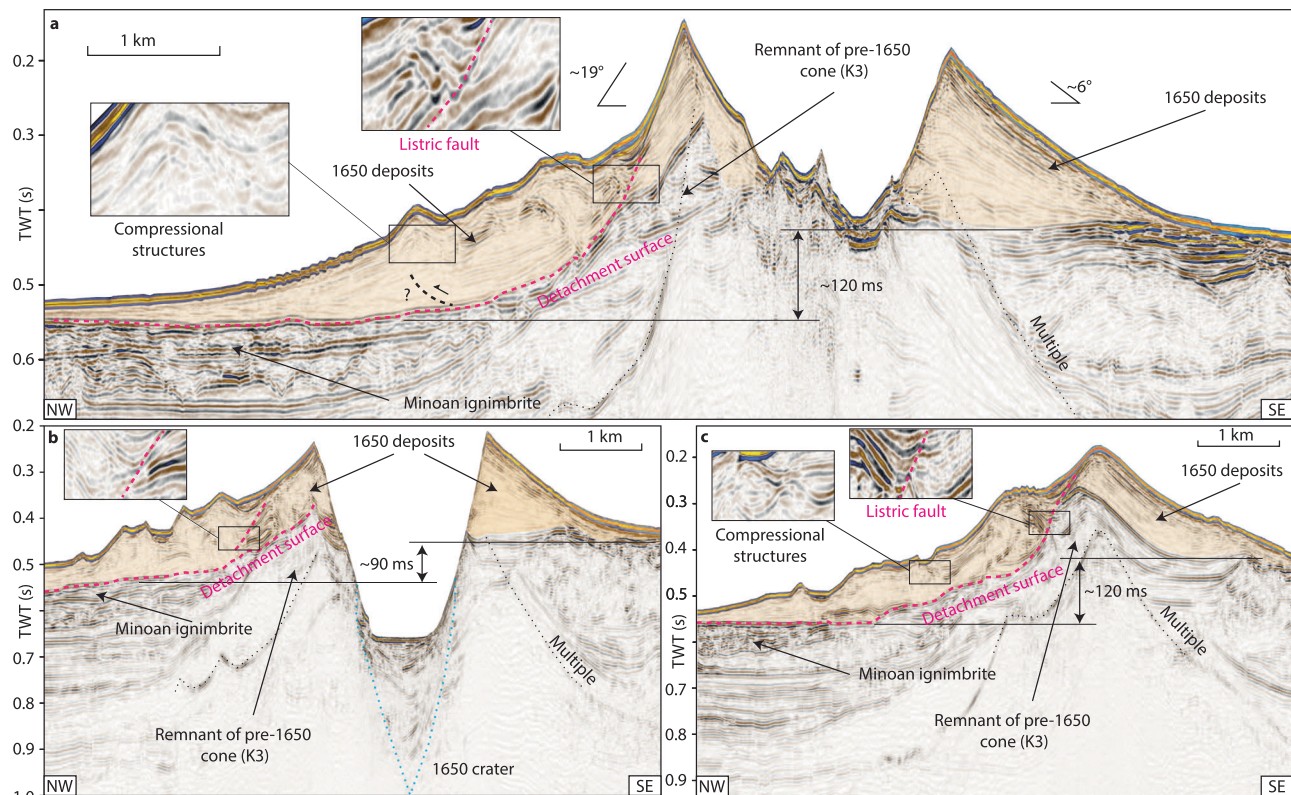

**Fig. 3 | Reflection seismic profiles of the Kolumbo edifice and flanks. a–c** Semi-transparent orange regions represent material deposited by the 1650 eruption. Folded and disrupted seismic reflections within the northwestern flank of the volcano are the result of internal deformation, while parallel reflections within the southeastern flank of the volcano indicate the absence of deformation. A tentative interpretation of a thrust beneath compressional folding is shown by the broken black line in (**a**), soling into the basal detachment surface (broken pink line). The locations of the profiles are shown in Fig. 1c.

2000 kg/m³. We compared the resulting flank deformation and tsunami heights with the geophysical observations and the eyewitness reports.

Our analyses reveal that a range of yield strength-density pairs can reproduce the observed tsunami run-up heights at Ios, Sikinos, and northern Santorini (Fig. 6a, Supplementary Figs. S1–S25). Given the deposit density range defined by the pumice sample analysis (1250–1500 kg/m³), it is possible to reproduce the tsunami observations with simulations using a yield strength between 5000 and 20,000 Pa, which is the same value range used in previous VolcFlow simulations of Anak Krakatau[29] and Ritter Island[27]. At Perissa and Kamari, the important observation of sea retreat is reproduced, but run-up heights are underestimated (blue dashed line in Fig. 6d–h).

In addition, we simulated an explosion-induced tsunami, which requires an estimate of the peak wave amplitude, previously assumed to be in the range of 150–240 m, to be able to reconstruct tsunami observations at Santorini[20]. However, considering the crater rim lies in an average water depth of only 135 m, the initial wave height by the explosion is unlikely to have exceeded this. Our simulations of an explosion-induced tsunami with a peak wave height of 150 m can neither reproduce the wave height observed at southern Ios and Sikinos nor the reported initial sea retreat at Kamari and Perissa (green dotted line in Fig. 6d–h).

Since the seismic data show that both source mechanisms could have contributed to tsunami genesis and neither the slope failure nor the explosion-induced tsunami simulations alone could reproduce eyewitness accounts, we performed additional model runs to simulate the effects of a combination of both source mechanisms, superimposing the wave field of an explosion onto that caused by flank deformation. The lag time between flank deformation and the explosion is unknown. However, we interpret that the explosion was caused by depressurization of a critical, erupting system or exposure of the feeder system, which would require lateral movement of the overlying sediment by a distance on the order of 500–1000 m. Given a simulated velocity of 3 to 4 m/s for the flank movement, this equates to a lag time of -2 to 5 min from the initiation of the slope instability. Our results show that assuming a lag time of 4 min from the beginning of flank deformation to the explosion, and an explosion-derived initial wave height of 150 m, we are able to simulate a tsunami pattern that is consistent with all known historical eyewitness accounts (red solid line in Fig. 6). In addition, various combinations of lag time and explosion-derived initial wave height show good agreement with the run-up heights at Ios as well as the initial sea retreat and subsequent run-up heights on the eastern Santorini coastline (Supplementary Figs. S1–S25). The observed inundation of 240 m on Sikinos cannot be directly transferred to a run-up height due to the simplified topography used in the simulations and the lack of a precise location of the eyewitness account. However, the reported inundation indicates that the southeastern coast of Sikinos was affected by a tsunami with significant run-up height. The good agreement between our simulations and the eyewitness accounts gives confidence in the combined simulation approach and highlights the complexity of tsunami generation related to the cascading chain of events during volcanic eruptions (Fig. 7). While a lateral explosion might be able to lead to a regional tsunami pattern consistent with historical eyewitness accounts, we do not have any evidence for such an explosion. We argue that the simplest explanation for tsunami genesis, supported by our data, is flank movement followed by explosive eruptions.

## Chain of events during the 1650 eruption
The combination of high-resolution marine geophysical datasets and historical eyewitness accounts offers a unique opportunity to

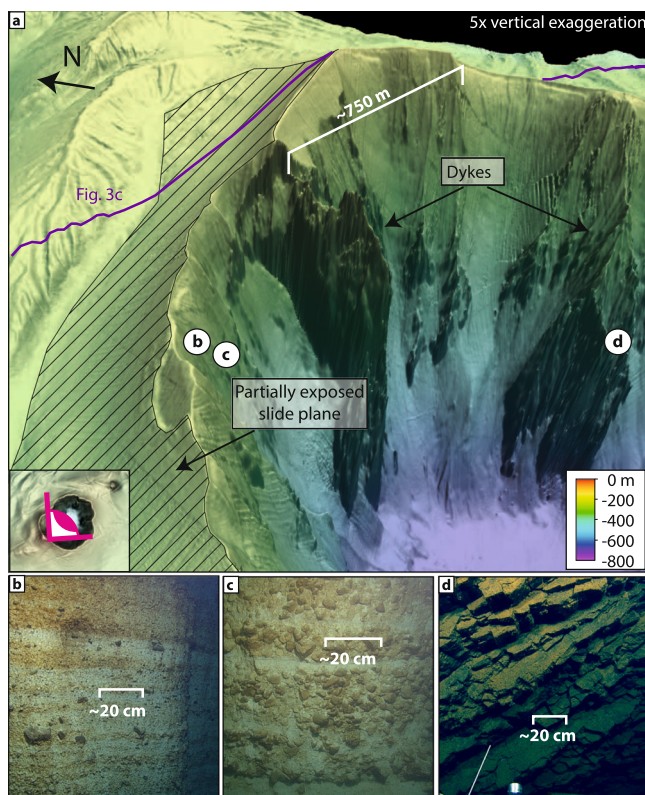

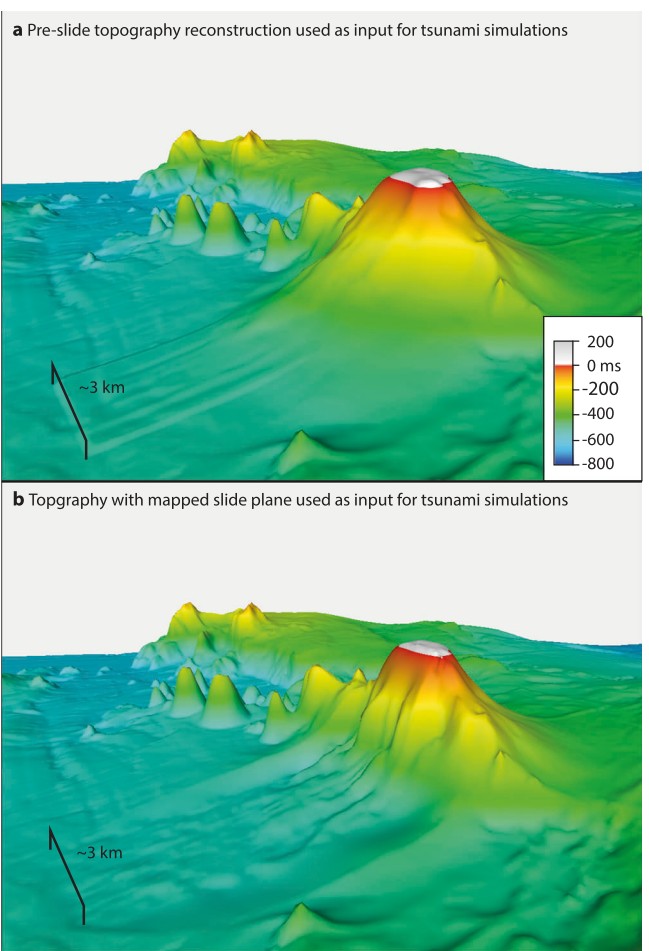

**Fig. 4 | Kolumbo's deformed edifice. a** 3D view of Kolumbo's northeastern crater wall with exposed dykes and the exposed detachment surface (slide plane). **b**, **c** ROV images of pumice layering representative of the 1650 cone's lithology. **d** ROV image of an exposed dyke within the crater.

**Fig. 5 | Cone and slide plane reconstructions used as simulation input. a** 3D view of the reconstructed 1650 eruption cone of Kolumbo. **b** 3D view showing the slide plane used for the numerical tsunami simulations.

reconstruct the sequence of volcanic processes that triggered the 1650 explosion and tsunami. The 1650 cone consists of highly vesicular pumice, which was deposited as fallout from the eruption column, where many of the large pumice clasts floated at the sea surface before becoming water-saturated and sinking[18] (Fig. 7a, b). The 1650 cone had a volume of approximately 4 to 6 km³ (ref. [16]), which was deposited in a very short timeframe of only 2 weeks, based on the onset of visually detectable eruption products in eyewitness accounts[18,20,21]. This rapid deposition has likely precluded significant consolidation of the pumice deposits covering the pre-1650 topography. The pre-1650 topography was characterized by the remnants of a cone on the northwestern side of Kolumbo from the previous eruptive cycle (K3 in Fig. 3), as well as a topographic step cutting through the volcanic edifice. Consequently, deposits from the 1650 eruption were deposited on steep slopes on the northwestern side of Kolumbo, while the southeastern side was relatively flat (Figs. 2 and 7b). ROV-dives targeting neighboring cones from the Kolumbo chain have revealed pronounced biogenic cementation of volcanic material[14,30], thereby forming a stark material contrast with the overlying 1650 eruption products, which is indicated by a positive polarity reflection (Figs. 2 and 3).

Slope stability is governed by sub-seafloor effective stress (total stress minus pore pressure) and material strength properties. The 1650 pumice deposits were sensitive to slope failure triggered by dynamic loading because (1) rapid deposition precluded the development of cohesion and (2) the exceptionally low deposit density (~1250–1500 kg/m³) resulted in low effective stress within the cone. We suggest that the pronounced material contrast between the cone and the older, underlying volcanic material, as well as the steeper northwestern slope (19° compared to ~6° elsewhere), explains why the failure occurred on the northwestern flank. Eruptive activity was accompanied by earthquakes noted in the eyewitness reports[18], which

will have caused ground acceleration and a likely associated dynamic pore pressure increase. This scenario is consistent with the development of a listric failure surface that cuts through the northwestern flank of the cone and soles out onto the pre-1650 material, defining the detachment surface of the slope instability (Figs. 2, 3 and 6c).

The water displacement caused by the movement of the volcanic flank not only triggered a tsunami, as demonstrated by our numerical simulations, but also directly affected the dynamics of the ongoing eruption. Considering that the active vent had emerged above sea level at this stage[18], failure of the northwestern flank probably caused the subaerial vent area to slide into the sea along with the flank. Based on the geometry of the detachment surface, the slope failure removed up to 200 m of material, thereby unroofing the underlying magmatic system. Assuming a bulk density of 1300–1500 kg/m³, this unroofing may have resulted in a pressure reduction of up to 2.6 to 3 MPa, affecting the underlying feeder system and magma reservoir at a depth as shallow as 2 km, as indicated by seismic full-waveform inversion results[31]. The failure will have exposed deeper levels in the vent directly to seawater, while crack formation and possible re-activation of existing failure planes will have allowed phreatomagmatic interactions within the edifice (Fig. 7d). Thus, both decompression through unroofing and interaction with seawater must have affected the upper parts of the magma feeding system within the timeframe of 4 min, as suggested by our model results. Decompression of the magma by unroofing will have led to an expansion of the magmatic fluid phase within the already critical system, directly followed by enhanced

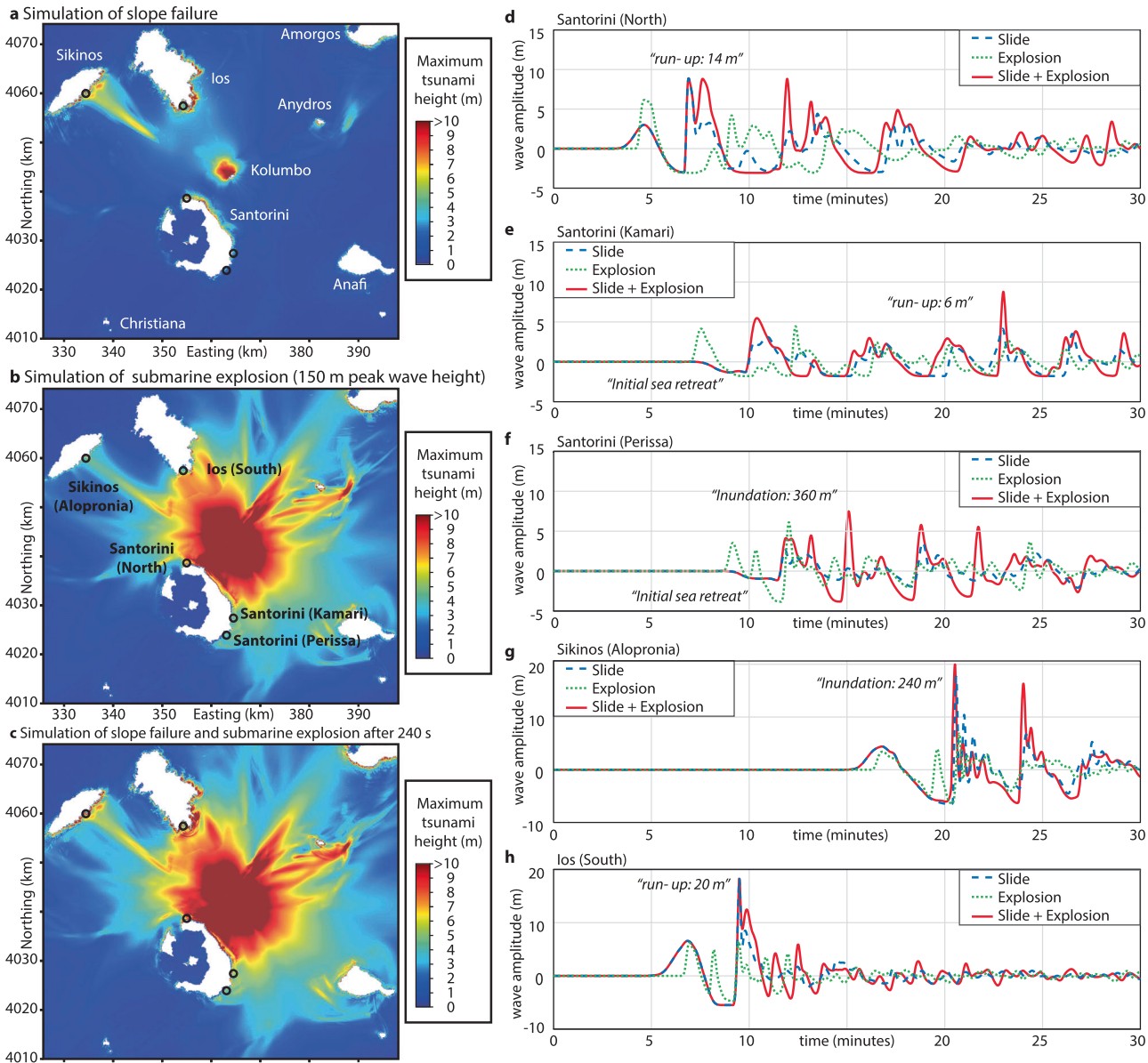

**Fig. 6 | Tsunami simulations of the 1650 flank deformation. a** Map showing the maximum tsunami heights of the simulation assuming a slide density of 1500 kg/m³ and a yield strength of 7500 Pa. **b** Map showing the maximum tsunami heights of the simulation assuming an explosion with a peak tsunami height of 150 m. **c** Map showing the maximum tsunami heights of the simulation assuming a slide density of 1500 kg/m³ and a yield strength of 7500 Pa and an additional explosion at 4 min after the initiation of the slide with a peak tsunami height of 150 m. **d–h** Virtual tide gauges showing the tsunami waveforms up to 30 min after the beginning of the flank deformation.

internal (closed-system) degassing and associated further pressure buildup.

The flank failure took place during an ongoing eruption, meaning that there was a hydraulic connection in the form of a highly vesicular magma[18] between the summit region and the deeper magma feeding system. A decompression of 3 MPa in the conduit at, e.g., 10 MPa ambient pressure (or about 600 m depth within the volcano base, assuming a magma density of 1500 kg/m³) would lead to a sudden volume increase of about 50% for bubbles existing in the magma, and thus produce efficient fragmentation in the upper vent region[32]. The pressure released upward in the open conduit would be sufficient to increase the internal bubble pressure above the typical yield strength of about 5 MPa for a felsic magma[33], leading to magma fragmentation at shallow conduit levels. The magma strain rate would suddenly exceed a critical limit[34], and efficient magma fragmentation would

commence at shallow levels. The increased fragmentation rates lead to a rapid downward migration of the fragmentation level, thereby initiating a chain reaction of accelerated decompression, producing sustained eruption rates, which further decompresses the deeper parts of the magma feeding system leading to a runaway process of magma ascent and subsequent depressurization. Decompression experiments[35] suggest a lag time of ~180 s between initial rapid decompression and accelerated degassing caused by bubble coalescence, which would be in agreement with the assumed slope failure dynamics. Magma-seawater interactions leading to phreatomagmatic explosions[36] must have played a central role at this stage, adding to the pressure buildup. The combined result of bubble expansion and phreatomagmatic explosions is rapid degassing in the magma feeding system, triggering a chain of highly explosive events and sustaining a runaway explosive eruption. Potential evidence of enhanced

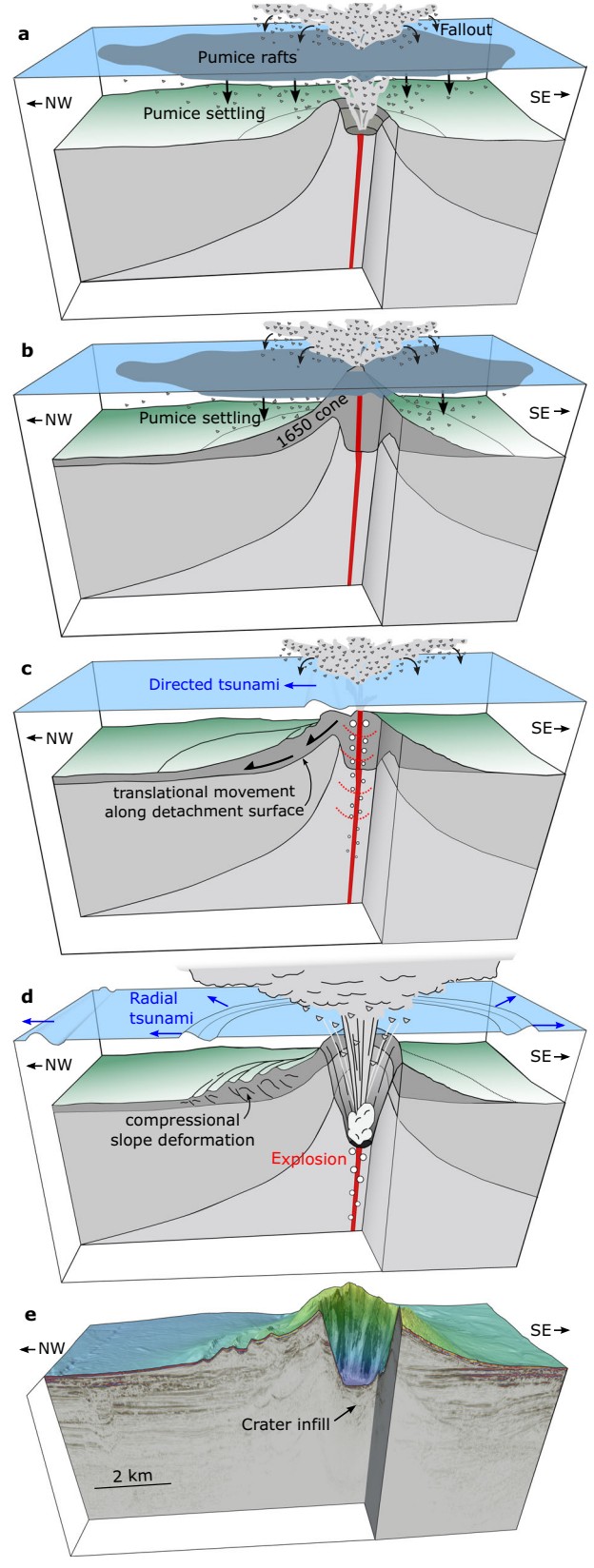

**Fig. 7 | Schematic of the Kolumbo volcanogenic tsunami cascade. a** Eruption initiates, producing large amounts of pumice that float on the sea surface before settling to the seafloor[3]. **b** Ongoing eruption and pumice settling results in the establishment of the 1650 cone, which breaches the sea surface[3]. **c** Northwestern flank deforms and moves downslope, causing a directed tsunami and resulting in depressurization of the underlying volcanic system. **d** Depressurization leads to degassing and expansion, which triggers a violent explosion. **e** High-resolution bathymetry and 3D seismic data reveal the present-day architecture of Kolumbo.

multiple caldera-forming eruptions at Tenerife[38], and initial results suggest that this process may have triggered the 2022 HTHH eruption[4]. The up to 2500-m-wide crater of Kolumbo is the result of one or multiple violent explosions, which agrees well with historical reports that the explosion was heard at least 400 km away at the Dardanelles[18]. The analysis of marine tephra in sediment cores[22] indicates that the 1650 eruption produced ~4.4 km³ (dense-rock equivalent (DRE) volume of ~2.5 km³) of tephra. Considering the large amounts of pumice forming the 1650 cone (~4 -6 km³, ref. [16]) and the reported pumice rafts floating throughout the southern Aegean Sea[18], the total eruption volume was larger than 10 km³ (5 km³ DRE). In comparison, the 1980 Mount St. Helens eruption produced only 1.1 km³ of tephra, while first mass flow rate-based estimates suggest an eruption volume of ~1.9 km³ for the 2022 HTHH eruption[39].

## Toward improved flank stability monitoring and volcanogenic tsunami early-warning systems

The 3D seismic analysis of Kolumbo highlights the complexity of tsunami genesis during volcanic eruptions, with flank deformation being the nucleus of a chain of events in a critical system. Internal deformation within volcanic flanks can progress over long periods, as seen by 3D seismic data of Ritter Island's failed flank[8] and pre-collapse InSAR data for Anak Krakatau[7]. The most prominent historical sector collapse triggered the 1980 eruption of Mount St. Helens, which was preceded by intrusion-controlled flank deformation. Although not all deformation of a volcanic flank will lead to a catastrophic collapse, deformation was a key precursor to these historical volcanic collapse events. Current monitoring strategies targeting the detection of flank deformation are based on satellite data and repeated photogrammetric imaging[7], methods that cannot be used to survey submerged volcanic edifices. Submarine geodetic networks have proven capable of detecting periodic slip events at Mount Etna[11], indicating that submarine deformation monitoring is possible. However, neither remote sensing-based subaerial approaches nor submarine geodetic monitoring techniques allow real-time observations of flank deformation. This may not be a problem for detecting prolonged deformation as a precursory warning signal but would be unsatisfactory in the case of rapidly developing systems like Kolumbo or for detecting and characterizing the transition from slow to fast (tsunamigenic) flank movement. It is not yet known whether unroofing by a slope failure played a key role in triggering the 2022 HHTH eruption, nor whether the edifice underwent longer-term (weeks to months) deformation prior to the eruption.

There are several active (sub) marine volcanoes that have been affected by repeated slope failures in the past but which are currently not monitored. Kick'em Jenny (Lesser Antilles) is highly active and has been affected by repeated small-scale cone collapses in recent decades[40]. While these collapses have not resulted in significant tsunami events in recent history, the shallow water depth of the active vent (-190 m), and its formation on the scar of a major sector collapse, highlight the potential hazard. Slope stability analyses and numerical simulations indicate a significant tsunami hazard associated with a potential sector collapse[41], which would be amplified further by a phase of explosive eruptive activity, as indicated by the 1650 Kolumbo and 2022 HHTH eruptions. Another candidate for future submarine

phreatomagmatic fragmentation of Kolumbo magma may lie in the abundance of very fine-grained tephra found in close proximity to the volcano in box cores and chemically correlated to the products of the 1650 event[37].

Depressurization by unroofing is known to have triggered violent explosive eruptions such as the 1980 Mount St. Helens eruption and

flank monitoring is the Myojinsho submarine volcano in Japan[42]. Its 1952–1953 explosive eruptions bore many similarities to the evolution of Kolumbo by forming a ~4 km diameter symmetrical volcaniclastic cone, which breached the sea surface before being destroyed by violent explosions[42]. While there is no evidence for past slope deformation in the available low-resolution bathymetry data, the steeply sloped cone (21°) formed on the northeastern rim of a larger pre-existing caldera. Renewed activity could result in a flank collapse of the volcaniclastic cone and depressurization of the underlying magmatic system, similar to the Kolumbo eruption, with the potential for the generation of much larger tsunamis than those caused by the 1952–1953 cone building activity. Finally, Kolumbo itself poses a significant hazard to the densely populated northern and western coasts of Santorini. A low P-wave velocity zone at 2 to 4 km depth indicates a shallow melt reservoir beneath Kolumbo, and repeated earthquake swarms may indicate fracturing associated with ascending melts[31,43]. Renewed volcanic activity or a strong earthquake may reactivate the deep-seated deformation of the northwestern flank, with the potential to trigger tsunami waves that would arrive at Santorini within 5 minutes (Fig. 6). Bottom waters in Kolumbo's crater are highly acidic[44], and even a minor slope failure could potentially trigger a limnic eruption releasing large amounts of $CO_2$ and other toxic trace gases like $SO_2$. Given that most fatalities from the 1650 eruption of Kolumbo were associated with toxic gases[18], this scenario deserves particular attention. However, currently available submarine monitoring systems require long lead-in times (often several years) and costly infrastructure, which would have precluded monitoring efforts even if subaerial precursors had been identified. Local populations, decision-makers and scientists are currently unprepared for the threats posed by submarine eruptions and slope failures, as has been demonstrated by the recent 2018 sector collapse of Anak Krakatau and the 2022 HHTH eruption. Therefore, new shore-line crossing monitoring strategies (as tests as part of the SANTORY project[45]) are required that are capable of being deployed as part of rapid response initiatives during volcanic unrest and which enable real-time observation of slope movement.

## Methods

### 3D seismic survey

The high-resolution P-Cable 3D seismic dataset was collected during research cruise POS538 aboard *R/V Poseidon* in October 2019 (ref. 46). The receiver array consisted of 16 streamers (each with 8 channels, spaced 1.5625 m apart) attached to a cross cable of ~200 m length. The source was a $45in^3$–$45in^3$ generator-injector airgun (harmonic mode), fired every 4 s, with a nominal ship speed of 3.5 knots through water. Processing included geometry definition, trace editing (removing dead and very noisy traces), static time corrections, anomalous amplitude attenuation, frequency-wavenumber (FK) filtering and trace interpolation. We then applied adaptive subtraction multiple suppression on common receiver gathers using a 1D pattern-matching algorithm. The resulting data were sorted to common midpoints on a grid with cell sizes of 3.125 m by 3.125 m, corrected for normal moveout using a water velocity of 1517 m/s, and then stacked. Gaps were filled via post-stack trace interpolation in both the inline and crossline directions. Signal to noise ratio was further improved with a post-stack 3D F-K coherency filter. We migrated the data with a constant velocity (1517 m/s) 3D Stolt migration, followed by residual 2D finite difference migrations applied in the inline and then crossline direction. We estimated velocities for the finite difference migrations via semblance analysis of long-offset 2D seismic data that crossed our 3D survey. The final data volume has a maximum spatial resolution of 3.125 m, and a maximum vertical resolution (half wavelength) near the seafloor of ~7 m, based on a dominant frequency of ~100 Hz.

### Numerical tsunami simulations

We used the numerical simulations code VolcFlow for all presented simulations (details in refs. 26–29,47). VolcFlow simulates the slide as a fluid and couples the interaction between the slide and the water based on a depth-averaged approximation on a topography-linked coordinate system. VolcFlow applies a plastic behavior with turbulence as rheology, which is primarily controlled by yield strength and density. The code uses general depth-averaged equations of mass and momentum conservation[43,48]. The slide was defined by reconstructing a pre-slide topography (Fig. 5), which was achieved by using the geometry of the remaining cone and constraints about the pre-slide geometry (i.e., that the cone breached the sea surface) and mapping the detachment surface in the 3D seismic datasets as slide plane, which had to be converted into the time domain using a seismic velocity of 1.7 km/s (ref. 15). The simulations had a lateral resolution of 100 m and simulated the wave field for 30 min. We performed 20 slide simulations using various densities (1250, 1500, 1750 and 2000 kg/m³) and yield strengths (5000, 7500, 10,000, 20,000 and 50,000 Pa) to test the parameter space (see. Supplementary Figs. S1–S25). To simulate the effects of an explosion, we followed the approach by Ulvrova et al.[20] and assumed a radial-symmetric initial explosion-induced water undulation for four different peak wave heights (100, 125, 150 and 240 m). Finally, we performed 20 simulations combining a slide and explosion using the same explosion-induced water undulations and five different time gaps between slide and explosion (1, 2, 3, 4, 5 min) combined with a slide with a representative density of 1500 kg/m³ and yield strength of 7500 Pa (as described in ref. 26). For this, the initial wave field from the explosion-derived tsunami simulations was added to the wave field of the landslide tsunami simulations at one specific time step (1, 2, 3, 4, 5 min) and the code used this superimposed wave fields for the rest of the simulations. To compare simulated wave heights with eyewitness accounts, we created maps showing the maximum wave height as well as virtual tide gauges at locations where direct tsunami observations are available. The virtual tide gauges were placed in shallow water (5–10 m and not at the coastline) to allow analyzing a potential sea retreat at each location.

### Autonomous underwater vehicle (AUV) mapping

During cruise POS510 (ref. 44), detailed bathymetric data were acquired through 7 planned dives by AUV Abyss, which is equipped with the dual frequency (200/400 kHz) RESON Seabat 7125 multibeam. During the acquisition, the nominal frequency of 200 kHz was applied, and all the logged data were further processed using the open-source software package MB-System. First, the navigation was adjusted using the interactive graphical interface program MBnavadjust and then merged with the bathymetric data. The following step included identifying and cleaning erroneous beams; therefore, dedicated filters were applied and manual cleaning through 2D and 3D editors. Finally, the processed bathymetric data were gridded, yielding a spatial resolution of 2 m, using the function mbgrid and, more specifically, the Gaussian weighted mean algorithm.

## Data availability

The AUV bathymetry dataset is available at the PANGAEA data repository [https://doi.org/10.1594/PANGAEA.958275]. The entire 3D seismic dataset will become available in September 2025 at the PANGAEA data repository [https://doi.org/10.1594/PANGAEA.960973]. For previous access on reasonable request, please contact the authors.

## Code availability

The used numerical tsunami simulation code VolcFlow can be downloaded at https://lmv.uca.fr/volcflow/.

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

## Acknowledgements

We thank the masters and crews of RV Poseidon for their support during research cruise POS538. We thank Mark Hannington and Sven Petersen for providing access to the AUV bathymetry dataset. Thanks also to Dirk Klaeschen for giving valuable advice at several key stages of seismic processing. We also thank Schlumberger and IHS for granting educational licenses to their software.

## Author contributions

J.K. and C.B. were responsible for organizing and designing the reflection seismic experiments. J.K., G.J.C., J.P., F.S., P.N., M.K., G.D.V., and J.E. acquired, processed and interpreted the reflection seismic data. S.C. provided the ROV footage from within the crater. J.K. and K.K. performed the numerical tsunami simulations. J.K., G.J.C., T.H., and S.C. integrated the geophysical results into the volcanological framework of Santorini. J.K., G.J.C. and T.H. drafted the manuscript, while J.P., S.C., P.N., M.K., G.D.V., F.S., K.K., J.E., and C.B. discussed the dataset and provided comments and corrections to the manuscript.

## Funding

## Competing interests

The authors declare no competing interests.
