## [Peer Review File · Nature Communications]

Cascading events during the 1650 tsunamigenic eruption of Kolumbo volcanoREVIEWER COMMENTS

Reviewer #1 (Remarks to the Author):

I read with great interest the paper by Jens Karstens and colleagues. The topic addressed is very interesting and useful for the scientific community and not only. Understanding the processes and mechanisms of volcanogenic tsunamis is of great importance for protecting coastal communities and for being prepared.

The authors combine high-resolution three-dimensional seismic data, bathymetric data and modeling to reconstruct the 1650 tsunamigenic eruption of Kolumbo volcano, near Santorini. They also use eyewitness data to assess their results.

In my view, the paper is very well organized, the methodological approach is clear, and the conclusions are supported by the data.

Reviewer #2 (Remarks to the Author):

This is a very interesting study about the Kolombo volcano that collapsed 470 years ago, triggering a tsunami in the Aegean Sea. The authors explore high resolution seismic data to study the internal structure, and develop a conceptual model of depressurized magma associated with the unloading event. The provided data includes bathymetric data, 3D seismic reflection profiles, and tsunami simulations, all of which of high quality and nicely demonstrated.

I enjoyed reading this work, but also have a number of suggestions that may improve the work. Importantly i ask the authors to more carefully choose the wording (often imprecise), to better use the data and develop a quantitative analysis (currently field data is mainly quantitative and descriptive), and to make the data publicly available. Once these and the moderate points listed below are considered, i believe the manuscript will make an excellent contribution and be well placed in ncomms.

Point listing:

- 1) The abstract is generally very vague and imprecise: i suggest complete rewriting, toning down provocative statements, and adding more detail on what actually was done, and add quantitative results if available.

- 2) L 23: how was the depressurization determined/quantified?
- 3) L 24: what means "notoriously unstable"? how determined? it is a very vague statement. I would expect rock mechanical tests to support this when reading this abstract.
- 4) L 27: ...mechanism followed by submarine explosive eruption. >> maybe it was simultaneously? Can the chicken and egg really be separated here?
- 5) L 28: You write " Our analysis represents the first integrated quantitative reconstruction of a cascading sequence of natural hazards". This obviously is wrong. I do not start listing citations of other cascading studies, but there are many integrated and quantitative reconstructions and monitored cascades, from earthquakes/landslides/liquefaction or from earthquake/volcano/landslide or other cascades. Please consider toning down your statements or expect a flood of critical comments after you are publishing this.
- 6) L 34: "While there is little known..." > this statement is about the eruption or the tsunami of HTHH? There are papers out on the HTHH now. Please implement and rewrite.
- 7) L 36: Replace "satellite altimetry" by "satellite geodetic"
- 8) L 39: "Internal deformation..." > can you add more details on how fast these displacements actually are at the cases provided? Are we talking about mm or m displacements? During which period?
- 9) L41-42: I would add here a sentence or two making clear where new research is demanded, or what is actually the added value of this paper. In actually most examples of volcano-induced tsunami, the search for the trigger mechanism remained debated. Even for the 1883 and again the 2018 Krakatau collapses, discussion persists weather the tsunami were triggered by the flank collapse, by a caldera, by explosions, by pyroclastic flows entering the sea, or by more than one of these hazards. Please better introduce this state of the art.
- 10) L 41: Here you refer to Mt Etna and Kilauea > for all these cases mentioned, cascading sequences have been proposed. If this is an important element of the paper (as proposed in your abstract), but this is not "the first integrated work".
- 11) L 44: should read "...350 ka ago"
- 12) Authors use inconsistent formats. "Year" is abbreviated on L 45 as "yrs", and just one line above on L 44 as "a".
- 13) L 45: 1650 of what? a.d.?
- 14) L 52: "Sikinos" > please add citations after this statement before the next statement follows.
- 15) L 56: 360 m > this is a horizontal measure?
- 16) L 58: previous studies also speculate about an interlinked mechanism
- 17) Structural architecture of Kolumbo > this chapter (the result chapter) is highly qualitative only and contains a mixture of descriptive terms, interpretation and discussion. I suggest to carefully check again how a quantitative analysis can be strengthened, where dimensions of blocks, thicknesses of layers, offsets of faults and deviations in strike are determined and investigated.

18) L 66. i am aware that the journals format has methods at the end of the work. But it is a bit difficult to follow if it is not mentioned before that here a new marine dataset is analysed showing the bathymetry and lithology variations associated with a submarine flank failure.

19) L 70-71: maybe it would be good to better explain how you can see this in figure 2. I presume that by following the traces of lithology you determine if a flank is deformed or undeformed? Currently the text reads like a rough interpretation, but lacks the scientific description.

20) L 75: "of $\sim 1.2 \text{ km}^3$ " > please add errors. Similar also for other estimates, such as the seismic velocity of 1700 m/s (same line) i would expect to see a more rigorous analysis. What are the method/data and analysis uncertainties? These are propagating into the volume errors, i presume.

21) L 77: "Detailed analysis of 3D seismic data..." > what is meant by that? You just zoom in an look closer, or you use any particular technique to improve the analysis? Please add some details here.

22) L 78: i am wondering how much subject interpretation is allowed in a "results" paragraph. Maybe the authors may want to reconsider the structure here, i usually prefer separation of data analysis and interpretation.

23) L 79: dont use "the fact". If i read this, i start being suspicious. Better is to provide solid data and arguments, and therewith to substantiate your statement.

24) L 80: "unequivocally" > seems we are here leaving scientific literature. Is that subjective feeling? The text is not getting more convincing with that type of expression.

25) L 81: "... not contourite sedimentation" > what is meant here? That it is not a continental rise? Consider rewording.

26) L 81: Bathymetry or Morphology is complex, which suggest deformation is partitioned by distinct blocks... or similar. Please consider rewording. The word "deformation" is used by geodesists in a different way.

27) L 82: why is it a decollement? this seems rather a headwall, with an unknown amount of erosion.

28) L 84: this slope of 19° is rather shallow, especially as tension faults at the surface and at other deep listric landslide expose much steeper elsewhere. Does this suggest the landslide to be translational rather than rotational? Then it should be not deep rooted. Please consider to add such an analysis / discussion.

29) L 90: Please explain why the morphological step (or kink in slope) "likely" controls the instability.

30) L 93-95: Can you really rule this out? Since HTHH we have learned that large explosions occur and trigger tsunamis without deposition of "sizeable pyroclastic flows", and that smaller local tsunami are triggered by very localized PDCs. Please phrase it more carefully.

31) L 98: is it the decollement or the amphitheater?

32) L 101: "obvious" > Please tone down, and choose a more careful wording.

33) L 113: Maybe these model constraints are unlikely. Explosions may also be directed or lateral, which was not tested by these previous modelling studies.

- 34) L 126: Are the "eyewitness reports" available? Add reference of those used in here.
- 35) L 134-140: i do not understand why an explosion-induced tsunami model is shown in the main text, although before it is stated that this is ruled out. Consider moving this to the supplement.
- 36) L 141: "Seismic data clearly show..." > delete "clearly". Readers want to be convinced by data and arguments, and not by subjective wordings.
- 37) L 146: "we attribute the explosion to depressurization of..." > i do not understand this. Consider rewording.
- 38) L 147: what is a lateral translation? A displacement? A dislocation? An impulse?
- 39) L 148: "deformation velocity of 3 to 4 m/s" > please check your terminology, also here it is wrong. Deformation refers to the change in size or shape of an object. Displacements are the absolute change in position of a point on the object. You refer to displacement velocity here. Here and at many other places in the manuscript; please check very carefully.
- 40) L 151: "we are able to reproduce all observations" > also here it is needed to more carefully phrase this. How complete are these observations, how was this preselected? Please tone down the language.
- 41) L 174: The effective stress is related to total stress and pore pressure. Is it this what the authors refer to?
- 42) L 177 and following: the discussion about material contrasts can be strengthened, especially as a number of new studies suggest that these contrasts can also evolve by hydrothermal alteration.
- 43) L 181: why the ground acceleration has to be "strong"?
- 44) L 181: why "ultimately"?
- 45) L 184: maybe i missed something here, but what exactly is meant by "water displacement"?
- 46) L 187-88: why it has to be instantaneous, and what is near-instantaneous. These phrases are all very descriptive, not scientific, and leave behind a rather mixed sceptical reader. Please carefully rewrite these sentences.
- 47) L 192: these 2.6 - 3 MPa assume lithostatic unloading, right?
- 48) L 197: an ambient pressure of 10 MPa would correspond to which depth and magmatic pressure?
- 49) L 198 and following. I recommend discussing this more openly. Not only bubble growth may follow. There may be also reactivations of failure planes, the Coulomb failure stress may change, and other effects may have played a role.
- 50) L 204: why eruption rates should increase? Please be careful to not end up in a purely speculative discussion, which seems undermining an otherwise really nice study and database.
- 51) L 206...: rapid, catastrophc, highly explosive, ... please tone down.
- 52) L 208: i do not understand why authors now speculate about bubble coalescence. Any support from the field?

53) L 246: which several active volcanoes are these?

54) L 254: eruptions bore similarities... > is this english? I am not a native speaker, but this sounds incorrect.

55) L 261: how large is the significance mathematically?

56) L 272-274: This is not fully correct. Years prior to the Anak Krakatau collapse, a scientific paper forecasted the direction, scale and threat of a sector collapse. The problem is more complex, where especially communication strategies and awareness is lacking. Please consider looking into the related literature before speculating.

57) L 372: 372 Materials & Correspondence. For data enquiries, please contact Jens Karstens. I think this is NOT appropriate nowadays anymore. Please upload bathymetric data and ROV images to supplementary or an accessible online database.

Figure 1: A) Figs should read Fig. B) hard to read and small text. C) add scales of elevation (or is it the same as in B)?), and make distance scale more visible. The label "Deformed flank" appears at a location of erosive gullies.

Figure 2: Very nice. Listric faults usually come with block wise and antithetic faults. Is the small graben structure showing this; or why authors highlight outward-dipping faults only?

Figure 3. Can the dimension of the dikes be estimated? How shure are authors this plane is the actual "sliding plane"? Can B, C and D images be located on the A map?

Figure 5: the block diagrams lack important structural features from this work, such as the decollement, surface faults and so on. I suggest to try having it less speculative, and more closely related to the actual findings of this manuscript.

Reviewer #3 (Remarks to the Author):

The paper provides a very interesting investigation of the 1650 Kolumbo tsunamigenic eruption. Starting from the analysis of high-resolution three-dimensional seismic data, the authors provide a novel hypothesis regarding the complex sequence of events that culminated in the 1650 explosion. The hypothesis is validated by simulating the ensuing tsunami starting from three different generation mechanisms and by comparing the results with the available historical accounts on the tsunami features (polarity of the first arrival and maximum wave amplitudes at five different sites/areas). The conclusion that the tsunami was likely generated by the combined effect of a submarine landslide along the northwestern flank of the volcano followed after few minutes by the explosion seems sound and well supported by data interpretation and tsunami simulations.

So, my opinion is that the paper deserves to be published.

I have nonetheless a few issues to raise, which I think could contribute to make the paper more complete.

As a tsunami modeller, I am especially interested in some modelling details. I must admit I am not familiar with VolcFlow, but I guess it shares some common features with other numerical codes.

- If applicable, what was the spatial resolution used for the numerical simulations?
- For the landslide simulation, what kind of rheology was adopted? In understand you specified density and yield strength, but this could apply to different rheologies.
- In the depth averaged equations, are non-linear and/or dispersion terms accounted for in this study?
- A delicate point regards the combined generation mechanism. The authors state: "[...] to simulate the effects of a combination of both source mechanisms, superimposing the wavefield of an explosion onto that caused by flank deformation.". Does this mean the two mechanisms were treated separately and then the results summed together to obtain the combined effect? If this is true, then I think this procedure is not appropriate for highly non-linear generation mechanisms like the ones considered. I ask the authors to comment on this.
- How are the coastal boundaries treated? From the inspection of the computed tide gauge-records, it looks like a minimum depth has been used, pointing at the idea that linear simulations were run. Please clarify, taking also into account that, especially for the explosion, the tsunami generation mechanism is expected to be highly non-linear (see also my previous point).

Moreover, I ask the authors to detail more the criterion adopted to establish the water elevation threshold to assess the consistency with the historical accounts. In particular, in Sikinos the historical information regards an inundation extent of 240 m: how is this converted into maximum water elevation? If linear tsunami simulations are run and here non inundation is modelled, where are the maximum elevation computed? Along the coastline with a minimum depth attached? In correspondence

with a selected isobath? Using observed run-up in the comparison with computed maximum water elevations could not be appropriate and may bias the consistency assessment. Can the authors be more detailed on this aspect?

Few typos and possible improvements are listed hereafter.

Row 52: "western coast of Santorini": is it "eastern coast of Santorini"?

Row 95: here the width of the crater is indicated in 2500 m (as in row 217), while at row 47 it is of 1500 m. I am a bit confused.

Row 131: "Krakatau" instead of "Krahtau".

Row 232: separate "whichwas".

Row 282: "Christiana" instead of "Christian"?

Figure 2: Correct "Remant" into "Remnant" in all three panels.

Row 3512: (1,2,3,4,5 min).

Supplement 2, Figures S5 - S26.

In the caption, it would be desirable to summarise very briefly the criteria used to assess the consistency with the historical accounts. Maybe you could add in each tide gauge record plot an horizontal line showing the imposed threshold in amplitude.

Best regards,

Alberto Armigliato

Rebuttal Letter

We would like to thank Alberto Armigliato and two anonymous reviewers for their helpful and constructive comments, which helped us to improve the quality and accessibility of the manuscript. Following the recommendations and advice of the reviewers:

- 1) We have updated the abstract following reviewer 2's advices, while shortening it significantly to fulfil the journal requirements
- 2) We have moved the figure showing the topographies used as input for the numerical simulations from the supplement to the main text.
- 3) We have added an additional figure showing a 3D view on the seismic dataset.
- 4) We have edited all figures according to journal requirements
- 5) We have added a detailed data availability statement (uploading/data management process is still in progress)

In the following, you can find a detailed point-to-point response (line numbers refer to the revised manuscript without track changes):

REVIEWER COMMENTS

Reviewer #1:

I read with great interest the paper by Jens Karstens and colleagues. The topic addressed is very interesting and useful for the scientific community and not only. Understanding the processes and mechanisms of volcanogenic tsunamis is of great importance for protecting coastal communities and for being prepared.

The authors combine high-resolution three-dimensional seismic data, bathymetric data and modeling to reconstruct the 1650 tsunamigenic eruption of Kolumbo volcano, near Santorini. They also use eyewitness data to assess their results.

In my view, the paper is very well organized, the methodological approach is clear, and the conclusions are supported by the data.

We appreciate this positive feedback.

Reviewer #2:

This is a very interesting study about the Kolombo volcano that collapsed 470 years ago, triggering a tsunami in the Aegean Sea. The authors explore high resolution seismic data to study the internal structure, and develop a conceptual model of depressurized magma associated with the unloading event. The provided data includes bathymetric data, 3D seismic reflection profiles, and tsunami simulations, all of which of high quality and nicely demonstrated.

I enjoyed reading this work, but also have a number of suggestions that may improve the work. Importantly i ask the authors to more carefully choose the wording (often imprecise), to better use the data and develop a quantitative analysis (currently field data is mainly quantitative and descriptive), and to make the data publicly available. Once these and the moderate points listed below are considered, i believe the manuscript will make an excellent contribution and be well placed in ncomms.

We appreciate the reviewer's positive feedback and the detailed and constructive comments and suggestions.

Comment 1: The abstract is generally very vague and imprecise: I suggest complete rewriting, toning down provocative statements, and adding more detail on what actually was done, and add quantitative results if available.

We acknowledge the reviewer's impression of the abstract. We have rewritten parts of the abstract, also to fulfil the journal's requirements for abstract length. We are satisfied that the revised abstract is a solid reflection of the research carried out and the most important findings:

Volcanic eruptions can trigger tsunamis, which may cause significant damage to coastal communities and infrastructure. Tsunami generation during volcanic eruptions is complex and often due to a combination of processes. The 1650 eruption of Kolumbo submarine volcano triggered a tsunami causing major destruction on surrounding islands in the Aegean Sea. However, the source mechanisms behind the tsunami have been disputed due to difficulties in sampling and imaging submarine volcanoes. Three-dimensional seismic data show that ~1.2 km³ of Kolumbo's north-western flank moved 500 – 1000 m downslope along a basal detachment surface. This movement is consistent with depressurization of the magma feeding system, causing a catastrophic explosion. Numerical tsunami simulations indicate that only the combination of flank movement followed by an explosive eruption can explain historic eyewitness accounts. This cascading sequence of natural hazards suggests that assessing submarine flank movements is critical for early warning of volcanogenic tsunamis.

Comment 2: L 23: how was the depressurization determined/quantified?

We determined this based on our reconstruction of the cone's pre-slide topography (see new figure 5), which is described in lines 202 to 206: **"Based on the geometry of the detachment surface, the slope failure removed up to 200 m of material from the underlying system. Assuming a bulk density of 1,300 to 1,500 kg/m³, this may have resulted in a pressure reduction of up to 2.6 to 3 MPa, affecting the underlying feeder system and magma reservoir at a depth as shallow as 2 km, as indicated by seismic full-waveform inversion³²."**

Comment 3: L 24: what means "notoriously unstable"? how determined? it is a very vague statement. I would expect rock mechanical tests to support this when reading this abstract.

We agree that this statement is too vague. We have removed the words, which were indeed superfluous.

Comment 4: L 27: ...mechanism followed by submarine explosive eruption. >> maybe it was simultaneously? Can the chicken and egg really be separated here?

Yes, we think we can separate this with the tsunami simulations. The observations of an initial sea retreat at eastern Santorini points towards this. If both had been triggered simultaneously, the tsunami signal of both processes (slide and explosion) would have superimposed resulting in a wave peak arriving first. Such a scenario is not supported by the sea retreat at eastern Santorini.

Comment 5: L 28: You write " Our analysis represents the first integrated quantitative reconstruction of a cascading sequence of natural hazards". This obviously is wrong. I do not start listing citations of other cascading studies, but there are many integrated and quantitative reconstructions and monitored cascades, from earthquakes/landslides/liqefaction or from earthquake/volcano/landslide or other cascades. Please consider toning down your statements or expect a flood of critical comments after you are publishing this.

We agree with this and we have rephrased this part of the abstract.

Comment 6: L 34: "While there is little known..." > this statement is about the eruption or the tsunami of HTHH? There are papers out on the HTHH now. Please implement and rewrite.

We agree and have rephrased the sentence in line 34:” **Both eruptions have become a focus site for volcanological research in recent years.**”

Comment 7: L 36: Replace "satellite altimetry" by "satellite geodetic"

We changed the text accordingly.

Comment 8: L 39: "Internal deformation..." > can you add more details on how fast these displacements actually are at the cases provided? Are we talking about mm or m displacements? During which period?

We added in lines 41 to 43:” **At Mt Etna, an average seaward motion of 3 to 5 mm per year is observed¹¹, while the southwestern flank of Kilauea shows both a transient cm-scale yearly seaward movement, as well as m-scale slip events accompanied by major earthquakes¹³.**”

Comment 9: L41-42: I would add here a sentence or two making clear where new research is demanded, or what is actually the added value of this paper. In actually most examples of volcano-induced tsunami, the search for the trigger mechanism remained debated. Even for the 1883 and again the 2018 Krakatau collapses, discussion persists weather the tsunami were triggered by the flank collapse, by a caldera, by explosions, by pyroclastic flows entering the sea, or by more than one of these hazards. Please better introduce this state of the art.

We have added the following sentences in lines 59 to 62: “**The genesis of volcanogenic tsunamis is often complex and may involve underwater explosions, earthquakes, caldera subsidence, pyroclastic density currents, flank failures or a combination of these processes²². Even for prominent and comparably recent events, such as the 1883 eruption of Krakatau, the tsunami source mechanisms are still debated.**”.

Comment 10: L 41: Here you refer to Mt Etna and Kilauea > for all these cases mentioned, cascading sequences have been proposed. If this is an important element of the paper (as proposed in your abstract), but this is not "the first integrated work".

We have removed this statement in the revised abstract.

Comment 11: L 44: should read "...350 ka ago"

We changed the text accordingly.

Comment 12: Authors use inconsistent formats. "Year" is abbreviated on L 45 as "yrs", and just one line above on L 44 as "a".

We use “**yrs**” now throughout.

Comment 13: L 45: 1650 of what? a.d.?

We added “**CE**”.

Comment 14: L 52: "Sikinos" > please add citations after this statement before the next statement follows.

We have added the reference to Ulvrova et al., 2016.

Comment 15: L 56: 360 m > this is a horizontal measure?

Yes, the "inland" indicates this.

Comment 16: L 58: previous studies also speculate about an interlinked mechanism

This is correct. The analyses themselves are not interlinked, which is the argument that we intended to make here. We have changed "proposed" to "tested" to be more precise now.

Comment 17: Structural architecture of Kolumbo > this chapter (the result chapter) is highly qualitative only and contains a mixture of descriptive terms, interpretation and discussion. I suggest to carefully check again how a quantitative analysis can be strengthened, where dimensions of blocks, thicknesses of layers, offsets of faults and deviations in strike are determined and investigated.

While we acknowledge the request to be more quantitative, we are not able to be quantitative in the way that is being requested and the scope of the study does not require this. We have been quantitative about the volume of deformed sediments, based on careful mapping of the top and base of the deformed material. We do not observe displaced blocks that can be measured, nor can we determine discrete offset along faults in the deformed region. Even if we could measure these things, they would not give us better constraints on the acceleration and velocity of flank deformation – the parameters that would be required for more thorough constraints of tsunami generation. Despite this, we think that the strength of our study lies in the fact that we can link flank deformation to tsunami generation, and thereby test historical observations of the tsunami impact at different locations.

Comment 18: L 66. i am aware that the journals format has methods at the end of the work. But it is a bit difficult to follow if it is not mentioned before that here a new marine dataset is analysed showing the bathymetry and lithology variations associated with a submarine flank failure.

Since this is the requirement of the journal, we can't address this comment.

Comment 19: L 70-71: maybe it would be good to better explain how you can see this in figure 2. I presume that by following the traces of lithology you determine if a flank is deformed or undeformed? Currently the text reads like a rough interpretation, but lacks the scientific description.

We determine deformation of the flank due to the pronounced folding of internal reflections. Such observations are typical indicators for sediment deformation during mass movements. See lines 75-76: "While the southeastern flank comprises undeformed, sub-parallel strata, the northwestern flank displays pronounced internal deformation (**Figs. 2 and 3a**)."

We have added a more detailed description to the figure caption of figure 2: "**Semi-transparent orange regions represent material deposited by the 1650 eruption. Folded and disrupted seismic reflections within the northwestern flank of the volcano are the result of internal deformation, while parallel reflections within the southeastern flank of the volcano indicate the absence of deformation. Tentative interpretation of a thrust beneath compressional folding is shown by the broken black line in (a), soling into the basal detachment surface (broken pink line).**"

Comment 20: L 75: "of ~1.2 km³" > please add errors. Similar also for other estimates, such as the seismic velocity of 1700 m/s (same line) i would expect to see a more rigorous analysis. What are the method/data and analysis uncertainties? These are propagating into the volume errors, i presume.

The seismic velocities were determined in a previous study (Supplement of Preine et al., 2022). We draw reference to this on Line 81. This analysis resulted in a velocity of 1,680 m/s for the shallowest unit of Kolumbo (K5) and 1,720 m/s for the cone from previous activity (K3). We now write “**1700 m/s (±50 m/s)**”. However, this has only minor impact on the volume calculation and thus the ~1.2 km³ is still valid.

Comment 21: L 77: "Detailed analysis of 3D seismic data..." > what is meant by that? You just zoom in an look closer, or you use any particular technique to improve the analysis? Please add some details here.

We removed “detailed”. The 3D analysis involves scrolling through very narrowly spaced seismic profiles, which enables us to make interpretation with higher confidence compared to having only a single 2D seismic profile available.

Comment 22: L 78: i am wondering how much subject interpretation is allowed in a "results" paragraph. Maybe the authors may want to reconsider the structure here, i usually prefer separation of data analysis and interpretation.

While we appreciate this point, and acknowledge that other journals ask for a stricter delineation of observations and interpretations, the general style of Nature Communications allows interpretative statements in the results section. This style can allow the paper to flow better.

Comment 23: L 79: dont use "the fact". If i read this, i start being suspicious. Better is to provide solid data and arguments, and therewith to substantiate your statement.

We agree and changed “fact” to “observation”.

Comment 24: L 80: "unequivocally" > seems we are here leaving scientific literature. Is that subjective feeling? The text is not getting more convincing with that type of expression.

We removed this statement.

Comment 25: L 81: "... not contourite sedimentation" > what is meant here? That it is not a continental rise? Consider rewording.

We changed the sentence to “**Their curved shapes in planform, their relative steepness, and interpreted internal thrusting, indicate that the ridges are the result of compression and not contourite sediment depositional processes seen at some other submarine volcanoes²⁵.**”

Comment 26: L 81: Bathymetry or Morphology is complex, which suggest deformation is partitioned by distinct blocks... or similar. Please consider rewording. The word "deformation" is used by geodesists in a different way.

While we acknowledge that the word “deformation” can be used by different groups in different contexts to mean different things, the word is very commonly used to describe changes in the geometry of reflections caused by movement (either extensional or compressional movements). It is a common descriptive term used in the study of submarine landslides / flank collapse processes.

Comment 27: L 82: why is it a decollement? this seems rather a headwall, with an unknown amount of erosion.

We are referring to the basal detachment surface that our data image in the upper parts of the deformed flank. We highlight this in Figure 1c and 3a. We are not talking about a headwall, which we agree can not be defined as a decollement. However, we agree that the word decollement is perhaps not the best term, since it is commonly used for a subduction interface. Therefore, we have opted to change the term to “**detachment surface**” and have updated the text and figures accordingly. We point clearly to this feature in the figures.

Comment 28: L 84: this slope of 19° is rather shallow, especially as tension faults at the surface and at other deep listric landslide expose much steeper elsewhere. Does this suggest the landslide to be translational rather than rotational? Then it should be not deep rooted. Please consider to add such an analysis / discussion.

Since we observe a relatively flat detachment surface, we do indeed think that the flank deformation is more toward the translational style of movement than rotational movement. We have added the annotation “**translational movement along detachment surface**” to the block diagram in Fig. 7c, to clarify how we think the failure took place.

Comment 29: L 90: Please explain why the morphological step (or kink in slope) "likely" controls the instability.

The morphological step was likely not the most important factor for the instability, but the steep flanks of the pre-1650 topography. We have changed the statement: “**The 1650 eruptive products were deposited on top of remnants of a cone from a previous eruption (K3 in ref. 15), which introduce structural heterogeneities within the northwestern flank of Kolumbo, in contrast to the relatively flat southeastern flank (Fig. 2). This complex pre-eruption topography, with a steep northwestern flank, likely controlled the instability of this flank segment as slope angle is a dominant factor for slope stability²⁶.**”

Comment 30: L 93-95: Can you really rule this out? Since HTHH we have learned that large explosions occur and trigger tsunamis without deposition of "sizeable pyroclastic flows", and that smaller local tsunami are triggered by very localized PDCs. Please phrase it more carefully.

We have toned this down and rephrased to: “**Therefore, we deem pyroclastic flow emplacement and caldera collapse as unlikely tsunami source mechanisms.**”

Comment 31: L 98: is it the decollement or the amphitheater?

We have changed the term to “**detachment surface**”, which is a suitable descriptive term for this observation.

Comment 32: L 101: "obvious" > Please tone down, and choose a more careful wording.

“Obvious” is an appropriate term here. One definition of the word is “easily perceived”. We would definitely argue that a volcanic explosion is an easily perceived tsunami source mechanism.

Comment 33: L 113: Maybe these model constraints are unlikely. Explosions may also be directed or lateral, which was not tested by these previous modelling studies.

We understand the point here. We do not have any indications for a significantly lateral explosion, while we do have strong evidence for flank deformation. That is why we are testing the tsunamigenic potential of the flank deformation in this study. We have added a sentence toward the end of the manuscript stating: **“While a lateral explosion might be able to lead to a regional tsunami pattern consistent with historical eyewitness accounts, we do not have any evidence for such an explosion. We argue that the simplest explanation for tsunami genesis, supported by our new data, is flank movement followed by explosive eruptions.”** In lines 167 - 170.

Comment 34: L 126: Are the "eyewitness reports" available? Add reference of those used in here.

The eyewitness reports have been documented in the studies (in lines 51 - 58) that we cite in the section “Reassessment of the 1650 tsunami”. Citations are provided in this section to the relevant literature where these observations are stated.

Comment 35: L 134-140: i do not understand why an explosion-induced tsunami model is shown in the main text, although before it is stated that this is ruled out. Consider moving this to the supplement.

Perhaps there is a slight misunderstanding here. We do not rule out the explosion source. Rather, we do not think that an explosion by itself can be the *primary mechanism* to explain all of the historical observations. Our seismic reflection observations and modelling results show that flank movement is most likely to have been the primary control on tsunami genesis. **We think that explosive eruptions contributed to the tsunami, as stated in the lines 156 – 161: “Our results show that assuming a lag time of 4 minutes from the beginning of flank deformation to the explosion, and an explosion-derived initial wave height of 150 m, we are able to simulate a tsunami pattern that is consistent with all known historical eyewitness accounts (red solid line in Fig. 6). In addition, various combinations of lag time and explosion-derived initial wave height show good agreement with the run-up heights at Ios as well as the initial sea retreat and subsequent run-up heights on the eastern Santorini coastline (Supplement).”**

Comment 36: L 141: "Seismic data clearly show..." > delete "clearly". Readers want to be convinced by data and arguments, and not by subjective wordings.

Agreed. We have removed the word **“clearly”**. In addition we changed **“high confidence”** to **“confidence”**.

Comment 37: L 146: "we attribute the explosion to depressurization of..." > i do not understand this. Consider rewording.

We have changed the wording to **“However, we interpret that the explosion was caused by depressurisation...”**

Comment 38: L 147: what is a lateral translation? A displacement? A dislocation? An impulse?

Agree that this part of the sentence was ambiguous. We have changed the wording to: **“... which would require lateral movement of the overlying sediment by a distance on the order of 500 - 1000 m.”**

Comment 39: L 148: "deformation velocity of 3 to 4 m/s" > please check your terminology, also here it is wrong. Deformation refers to the change in size or shape of an object. Displacements are the absolute change in position of a point on the object. You refer to displacement velocity here. Here and at many other places in the manuscript; please check very carefully.

Agreed. We have changed the wording to **"Given a simulated velocity of 3 to 4 m/s for the flank movement, ..."**

Comment 40: L 151: "we are able to reproduce all observations" > also here it is needed to more carefully phrase this. How complete are these observations, how was this preselected? Please tone down the language.

To acknowledge that there is much that we don't know about the tsunami, and that we can only base our understanding on the eyewitness reports that have been written down, we have changed the wording here slightly to: **"Our results show that assuming a lag time of 4 minutes from the beginning of flank deformation to the explosion, and an explosion-derived initial wave height of 150 m, we are able to simulate a tsunami pattern that is consistent with all known historical eyewitness accounts."**

Comment 41: L 174: The effective stress is related to total stress and pore pressure. Is it this what the authors refer to?

Yes, that is what we are referring to. We have changed the wording slightly to: **"Slope stability is governed by sub-seafloor effective stress (total stress minus pore pressure) and material strength properties."**

Comment 42: L 177 and following: the discussion about material contrasts can be strengthened, especially as a number of new studies suggest that these contrasts can also evolve by hydrothermal alteration.

Considering that the 1650 Cone has been formed in a relatively short time interval, it appears unlikely (although not impossible) that hydrothermal alteration may have contributed to the slope failure. We would be happy to integrate this into the discussion references if the reviewer would point to specific publications.

Comment 43: L 181: why the ground acceleration has to be "strong"?

"Strong" is a qualifier that we cannot quantify. As such, we agree that the word is not useful. We have removed it and now just state **"ground acceleration"**.

Comment 44: L 181: why "ultimately"?

We have removed the word **"Ultimately"**.

Comment 45: L 184: maybe i missed something here, but what exactly is meant by "water displacement"?

Agree that the start of this sentence is not very clear. We have reworded it to: **"The water displacement caused by movement of the volcanic flank not only triggered a tsunami, ..."**

Comment 46: L 187-88: why it has to be instantaneous, and what is near-instantaneous. These phrases are all very descriptive, not scientific, and leave behind a rather mixed sceptical reader. Please carefully rewrite these sentences.

This is a fair point. We have rewritten these sentences and removed the word “instantaneous”.

Comment 47: L 192: these 2.6 - 3 MPa assume lithostatic unloading, right?

Correct – this assumes unloading of the highly porous and water saturated pumice material. The density we use for the pressure calculation here is the water-saturated density of the sediment.

Comment 48: L 197: an ambient pressure of 10 MPa would correspond to which depth and magmatic pressure?

Fair comment – We have included such information in the revised text: “...**10 MPa ambient pressure (or about 600 m depth within the volcano base, assuming a magma density of 1,500 kg/m³)...**”

Comment 49: L 198 and following. I recommend discussing this more openly. Not only bubble growth may follow. There may be also reactivations of failure planes, the Coulomb failure stress may change, and other effects may have played a role.

We have followed this suggestion and discussed the scenario in more detail, and now mention failure plane reactivation as an additional plausible cause for increased phreatomagmatic activity. We think that detailed descriptions of Coulomb failure are beyond the scope of the paper: “**The water displacement caused by movement of the volcanic flank not only triggered a tsunami, as demonstrated by our numerical simulations, but also directly affected the dynamics of the ongoing eruption. Considering that the active vent had emerged above sea level at this stage¹⁸, failure of the northwestern flank probably caused the subaerial vent area to slide into the sea along with the flank. Based on the geometry of the detachment surface, the slope failure removed up to 200 m of material, thereby unroofing the underlying magmatic system. Assuming a bulk density of 1,300 to 1,500 kg/m³, this unroofing may have resulted in a pressure reduction of up to 2.6 to 3 MPa, affecting the underlying feeder system and magma reservoir at a depth as shallow as 2 km, as indicated by seismic full-waveform inversion results³². The failure will have exposed deeper levels in the vent directly to sea-water, while crack formation and possible re-activation of existing failure planes will have allowed phreatomagmatic interactions within the edifice (Fig. 7d). Thus, both decompression through unroofing and interaction with seawater must have affected the upper parts of the magma feeding system within the timeframe of four minutes, as suggested by our model results. Decompression of the magma by unroofing will have led to expansion of the magmatic fluid phase within the already critical system, directly followed by enhanced internal (closed-system) degassing and associated further pressure buildup.**”

Comment 50: L 204: why eruption rates should increase? Please be careful to not end up in a purely speculative discussion, which seems undermining an otherwise really nice study and database.

This is a thoughtful comment. We agree that increased eruption rates may be speculative, and have omitted this expression.

Comment 51: L 206...: rapid, catastrophic, highly explosive, ... please tone down.

We have expanded the respective section and provided more detail, and thereby slightly toned down parts of the discussion. We agree with removing the word “catastrophic”, but think the phrase “highly

explosive” is appropriate in this context, in line with published data and observations on the Kolumbo 1650 eruption.

Comment 52: L 208: i do not understand why authors now speculate about bubble coalescence. Any support from the field?

We have provided more context to this section, in order to constrain the timescales involved in the triggering of the highly explosive eruption phase. Concerning field evidence, we now refer to the very high vesicularity of the pumice in the cone, and to the occurrence of fine grained Kolumbo ash in nearby surface sediments, both being evidence of very efficient fragmentation during the explosive events. In line with these observations, we cite laboratory decompression experiments of vesiculated melts, which show that bubble coalescence is an important pre-requisite for the triggering of efficient syn-eruptive magma fragmentation. We think the timescales involved in such bubble coalescence processes fit well with the suggested overall scenario, and provide complementary evidence for the timescales involved. With due respect, we cannot see that this line of evidence is speculative in the given context.

Comment 53: L 246: which several active volcanoes are these?

The text that directly follows this opening statement details examples of such active volcanoes: Kick'em Jenny, HTHH, Myojinsho.

Comment 54: L 254: eruptions bore similarities... > is this english? I am not a native speaker, but this sounds incorrect.

Yes, we checked it and it's correct English.

Comment 55: L 261: how large is the significance mathematically?

We are not talking about mathematical / statistical significance here. We are using the word “significant” to make clear that it is not a minor / insignificant hazard.

Comment 56: L 272-274: This is not fully correct. Years prior to the Anak Krakatau collapse, a scientific paper forecasted the direction, scale and threat of a sector collapse. The problem is more complex, where especially communication strategies and awareness is lacking. Please consider looking into the related literature before speculating.

We are aware of the scientific literature surrounding Anak Krakatau. We completely agree that the problem is complex and requires effective communication and awareness. We also think that monitoring strategies are required to be able to respond better to emerging hazards.

If we understand your point here correctly, we think you are not in agreement with our inclusion of “scientists” as a group that is unprepared. We think that scientists need to do more (i.e. monitoring) to be better prepared for future emerging hazards. However, to simplify this whole sentence, and tone it down a bit, we have changed the wording of the last two sentences to:

"Local populations are currently under-prepared for the threats posed by submarine eruptions and slope failures, as has been demonstrated by the recent 2018 sector collapse of Anak Krakatau and the 2022 HHTH eruption. We propose that new shore-line crossing monitoring strategies are required that are capable of being deployed as part of rapid response initiatives during volcanic unrest, and which enable real-time observation of slope movement."

Comment 57: L 372: 372 Materials & Correspondence. For data enquiries, please contact Jens Karstens. I think this is NOT appropriate nowadays anymore. Please upload bathymetric data and ROV images to supplementary or an accessible online database.

We have updated the data availability statement. We have uploaded the presented seismic profiles and the AUV bathymetry to repositories. The uploading process for the latter is still in progress. The full 3D seismic dataset is currently in the process of being uploaded and will be available in two years.

Comment 58: Figure 1: A) Figs should read Fig. B) hard to read and small text. C) add scales of elevation (or is it the same as in B)?, and make distance scale more visible. The label "Deformed flank" appears at a location of erosive gullies.

We have updated the figure accordingly

Comment 59: Figure 2: Very nice. Listric faults usually come with block wise and antithetic faults. Is the small graben structure showing this; or why authors highlight outward-dipping faults only?

This is a good point, and something we gave thought to during our interpretation of the deformation of the flank. However, after careful examination of the data, it became clear that we were not able to map out many individual faults with sufficient certainty between the detachment surface and the seafloor. Rather, the data reveal distributed deformation, mainly in the form of internal folding and seafloor steps, which are the basis for our interpretation of compressional deformation of the flank. We think we would risk “overinterpreting” the data if we attempt to speculate about antithetic faults and particular graben structures. Rather, we prefer to take a broader view of the large-scale deformation of the flank.

Comment 60: Figure 3. Can the dimension of the dikes be estimated? How sure are authors this plane is the actual "sliding plane"? Can B, C and D images be located on the A map?

The interpretation of the sliding plane bases on our seismic interpretations. However, since the slide plane is partially covered by material (see 2c), we have changed the label to “**Partially exposed slide plane**” in the figure labels. We have added the locations of the photographs to the figure. Based on ROV video footage, the dykes have widths of 5 – 10 m.

Comment 61: Figure 5: the block diagrams lack important structural features from this work, such as the decollement, surface faults and so on. I suggest to try having it less speculative, and more closely related to the actual findings of this manuscript.

We have added some more descriptive terms to the block diagrams to label the detachment surface and compressional deformation between the detachment and the seafloor. This links this conceptual diagram better to the observations shown from seismic data earlier in the manuscript.

Reviewer #3 (Remarks to the Author):

The paper provides a very interesting investigation of the 1650 Kolumbo tsunamigenic eruption. Starting from the analysis of high-resolution three-dimensional seismic data, the authors provide a novel hypothesis regarding the complex sequence of events that culminated in the 1650 explosion. The hypothesis is validated by simulating the ensuing tsunami starting from three different generation mechanisms and by comparing the results with the available historical accounts on the tsunami features (polarity of the first arrival and maximum wave amplitudes at five different sites/areas). The conclusion

that the tsunami was likely generated by the combined effect of a submarine landslide along the north-western flank of the volcano followed after few minutes by the explosion seems sound and well supported by data interpretation and tsunami simulations.

So, my opinion is that the paper deserves to be published. I have nonetheless a few issues to raise, which I think could contribute to make the paper more complete.

We would like to thank the reviewer for the helpful comments and suggestions.

As a tsunami modeller, I am especially interested in some modelling details. I must admit I am not familiar with VolcFlow, but I guess it shares some common features with other numerical codes.

Comment 62: If applicable, what was the spatial resolution used for the numerical simulations?

Our simulations were performed with a lateral resolution of 100 m. We have updated the description of the simulations and the used parameters in the method section:” **The tsunami simulations were performed with numerical simulations code VolcFlow (details in refs. refs. 27 – 30, 48). VolcFlow simulates the slide as a fluid and couples the interaction between the slide and the water based on a depth-averaged approximation on a topography-linked coordinate system. Volcflow applies a plastic behaviour with turbulence as rheology, which is primarily controlled by yield strength and density. The code uses general depth-averaged equations of mass and momentum conservation^{22,44}. The slide was defined by reconstructing a pre-slide topography (see Supplement S1), which was achieved by using the geometry of the remaining cone and constraints about the pre-slide geometry (i.e., that the cone breached the sea-surface) and mapping the detachment surface in the 3D seismic datasets as slide plane, which had to be converted into the time domain using a seismic velocity of 1.7 km/s; ref. 14). The simulations had a lateral resolution of 100 m and a simulation duration of 30 minutes. We performed 20 slide simulations using various density (1,250, 1,500, 1,750 and 2,000 kg/m³) and yield strengths (5,000, 7,500, 10,000, 20,000 and 50,000 Pa) to test the parameter space (see. Supplement 1). To simulate the effects of an explosion, we followed the approach by Ulvrova et al. (2016) and assumed a radial-symmetric initial explosion-induced water undulation for four different peak wave heights (100, 125, 150 and 240 m). Finally, we performed 20 simulations combining a slide and explosion using the same explosion-induced water undulations and five different time gaps between slide and explosion (1, 2, 3, 4, 5 minutes) combined with a slide with representative density of 1,500 kg/m³ and a yield strength of 7,500 Pa (as described in ref. 23). For this, the initial wave field from the explosion-derived tsunami simulations was added to the wave field of the landslide tsunami simulations at one specific time step (1, 2, 3, 4, 5 minutes) and the code used this superimposed wave fields for the rest of the simulations. To compare simulated wave heights with eyewitness accounts, we created maps showing the maximum wave height as well as virtual tide gauges at locations, where direct tsunami observations are available. The virtual tide gauges were placed in shallow water (5-10 m and not at the coast-line) to allow analysing a potential sea retreat at each location.”.**

Comment 63: For the landslide simulation, what kind of rheology was adopted? In understand you specified density and yield strength, but this could apply to different rheologies.

Volcflow applies a plastic behaviour with turbulence. We have added a statement to the method section (See comment 62).

Comment 64: In the depth averaged equations, are non-linear and/or dispersion terms accounted for in this study?

Volcflow does not account for dispersion. While dispersion may have a major effect for some wave field simulations, it has only a minor effect for our simulations as the wavelength of the tsunami is large compared to the relatively shallow water depth in the study area and as the tsunami source is close to the analysed coastlines.

Comment 65: A delicate point regards the combined generation mechanism. The authors state: "[...] to simulate the effects of a combination of both source mechanisms, superimposing the wavefield of an explosion onto that caused by flank deformation.". Does this mean the two mechanisms were treated separately and then the results summed together to obtain the combined effect? If this is true, then I think this procedure is not appropriate for highly non-linear generation mechanisms like the ones considered. I ask the authors to comment on this.

Yes, this is an important point. While tsunami generation by the sliding mass is a continuous, long-term (several minutes) process, we consider the generation by the explosion as (quasi) instantaneous. We defined the initial water undulation by the explosion following the approximation by Ulvrova et al., 2016. This wave field was added to the wave field of the landslide tsunami simulations at one specific time step (e.g. after 240 seconds like in figure 4c), which we believe is justified by the assumed instantaneous nature of tsunami generation by the explosion. The code then continue using this superimposed wave field for the rest of the simulations. Therefore, our combined simulations are not the sum of the wave field of the slide and that of the explosion at every time step.

Comment 66: How are the coastal boundaries treated? From the inspection of the computed tide gauge-records, it looks like a minimum depth has been used, pointing at the idea that linear simulations were run. Please clarify, taking also into account that, especially for the explosion, the tsunami generation mechanism is expected to be highly non-linear (see also my previous point). Moreover, I ask the authors to detail more the criterion adopted to establish the water elevation threshold to assess the consistency with the historical accounts. In particular, in Sikinos the historical information regards an inundation extent of 240 m: how is this converted into maximum water elevation? If linear tsunami simulations are run and here non inundation is modelled, where are the maximum elevation computed? Along the coastline with a minimum depth attached? In correspondence with a selected isobath? Using observed run-up in the comparison with computed maximum water elevations could not be appropriate and my bias the consistency assessment. Can the authors be more detailed on this aspect?

We have added a statement to the method section **“The virtual tide gauges were placed in shallow water (5-10 m and not at the coastline) to allow analysing a potential sea retreat at each location.”**. As mentioned, the explosion triggered tsunami could only be approximated using the approach by Ulvrova et al., 2016, which does not reflect the non-linearity of the explosion. The Volcflow simulation allow inundation of the tsunami wave. The criticism regarding including the eyewitness accounts from Sikinos into the analyses are fair considering the lateral resolution of only 100 m and the missing information, where exactly on Sikinos the water inundated for 240 m. While we still believe that the observation of a significant inundation is worth to be included in the manuscript, we have updated the manuscript indicating that that the inundation observation cannot be transferred into tsunami heights considering the limitations of the simulations: **“The observed inundation of 240 m on Sikinos cannot directly transferred to a run-up height due to simplified topography used in the simulations and the lack of precise location of the eyewitness account, but indicates that the southeastern coast of Sikinos was affected by a tsunami with significant run-up height.”**

Few typos and possible improvements are listed hereafter.

Comment 67: Row 52: "western coast of Santorini": is it "eastern coast of Santorini"?

Yes, this correct. Thank you for spotting this. We changed the manuscript accordingly

Comment 68: Row 95: here the width of the crater is indicated in 2500 m (as in two 217), while at row 47 it is of 1500 m. I am a bit confused.

It must be 2500 m. We have changed the manuscript accordingly.

Comment 69: Row 131: "Krakatau" instead of "Krahtau".

We have changed the manuscript accordingly.

Comment 70: Row 232: separate "whichwas".

We have changed the manuscript accordingly.

Comment 71: Row 282: "Christiana" instead of "Christian"?

We have changed the manuscript accordingly.

Comment 72: Figure 2: Correct "Remant" into "Remnant" in all three panels.

Thank you for spotting this. We have changed the manuscript accordingly.

Comment 73: Row 3512: (1,2,3,4,5 min).

We have changed the manuscript accordingly.

Comment 74: Supplement 2, Figures S5 - S26. In the caption, it would be desirable to summarise very briefly the criteria used to assess the consistency with the historical accounts. Maybe you could add in each tide gauge record plot an horizontal line showing the imposed threshold in amplitude.

This is good idea. We have added following statement to the figure captions: **“Threshold tsunami heights from eyewitness accounts used to evaluate the simulation results: 7.5 m at northern Santorini, 5 m at Perissa, 5 m Kamari, 10 m at southern Ios and 10 m at Sikinos. A direct transfer of inundation observations to tsunami height for Sikinos is not possible and thus the Sikinos threshold has is less reliable compared to the direct tsunami height information available for the other locations.”**

REVIEWERS' COMMENTS

Reviewer #2 (Remarks to the Author):

I have read the revised version of the manuscript and also the rebuttal letter provided. Looking at these documents provided, i see that all points I raised have been addressed thoroughly.

Reviewer #3 (Remarks to the Author):

I think the authors did a good job in revising the manuscript. All my comments and suggestions have been properly taken into account.

I send only few suggestions for typo corrections and slight modifications.

Page 1, row 40: correct "oberved" into "observed"

Page 8, caption to Figure 1. Please add a sentence like: "The cyan, red and purple lines indicate the position of the profiles depicted in Fig. 3."

Page 10, caption to Figure 3. Please add a sentence like: "The position of the three profiles is drawn in Fig. 1."

Page 12, Figure 5, panel a). Correct "reconstrucation" into "reconstruction". Panel b): correct "Topgraphy" into "Topography".

Page 15, rows 369-370: modify the sentence as "The tsunami simulations were performed with the numerical code VolcFlow ..."

Page 15, row 378: delete "s simulations"

Page 15, row 379: change “density” into “densities”

Page 20: the last reference appears to be truncated and incomplete.

Best regards,

Alberto Armigliato